# Assessing the impact of CHARM2, a family planning program on gender attitudes, intimate partner violence, reproductive coercion, and marital quality in India

Sangeeta Chatterji[1]*, Nicole E. Johns[2], Mohan Ghule[2], Shahina Begum[3], Sarah Averbach[2,4], Madhusudana Battala[5], Anita Raj[2,6,7]

1 School of Social and Political Science, University of Edinburgh, Edinburgh, United Kingdom, 2 Center on Gender Equity and Health, School of Medicine, University of California San Diego, San Diego, California, United States of America, 3 ICMR-National Institute for Research in Reproductive and Child Health, Mumbai, India, 4 Department of Obstetrics, Gynecology and Reproductive Sciences, School of Medicine, University of California San Diego, San Diego, California, United States of America, 5 Population Council, New Delhi, India, 6 Newcomb Institute, Tulane University, New Orleans, Louisiana, United States of America, 7 School of Public Health and Tropical Medicine, Tulane University, New Orleans, Louisiana, United States of America

* s.chatterji@ed.ac.uk

**Data Availability Statement:** Data and analysis code have been uploaded as part of this submission.

## Abstract

Using a two-armed cluster randomised controlled trial, CHARM2 (Counselling Husbands to Achieve Reproductive health and Marital equity), a 5-session gender equity and family planning intervention for couples in rural India, showed an impact on family planning outcomes in primary trial analyses. This study examines its effects on gender-equitable attitudes, intimate partner violence, reproductive coercion, and marital quality. We used multilevel mixed-effects models to assess the intervention impact on each outcome. Both male (aIRR at 9 months: 0.64, C.I.: 0.45,0.90; aIRR at 18 months: 0.25, C.I.: 0.18,0.39) and female (aIRR at 9 months: 0.57, C.I.: 0.46,0.71; aIRR at 18 months: 0.38, C.I.: 0.23,0.61) intervention participants were less likely than corresponding control participants to endorse attitudes accepting physical IPV at 9- and 18-month follow-ups. Men in the intervention, compared to those in the control condition, reported more gender-equitable attitudes at 9- and 18 months (ß at 9 months: 0.13, C.I.: 0.06,0.20; ß at 18 months: 0.26, C.I.: 0.19,0.34) and higher marital quality at the 18-month follow-up (ß: 0.03, C.I.: 0.01,0.05). However, we found no effects on women's experiences of IPV, reproductive coercion, or marital quality. CHARM2 shows promise in improving men's and women's attitudes towards gender equality and male perceptions of marital quality. Still, IPV and reproductive coercion reductions may require more intensive programming than that provided within this 5-session model focused on family planning.

**Funding:** This work was supported by the National Institutes of Health (R01HD084453 to AR) and the Bill and Melinda Gates Foundation (INV-002967 to AR). The funders had no role in study design, data collection and analysis, decision to publish, or preparation of the manuscript.

**Competing interests:** The authors have declared that no competing interests exist.

## Introduction

Family planning and reproductive freedom are central to women's health and well-being. However, the uptake of modern contraception is still low in India. In 2019–21, 67% of married women 15–49 years used any family planning method, and 57% used modern contraception [1]. Women have to overcome several gender equity and power-related barriers to achieve reproductive freedom, including male control over women's reproductive health. Men's gender attitudes and engagement in family planning can play a crucial role in improving women's health outcomes in low- and middle-income countries (LMICs) [2–4]. In patriarchal societies like India, men can often be the chief decision-makers and influence women's access to reproductive and maternal healthcare. Intimate partner violence (IPV) and reproductive coercion (RC), both forms of gender-based violence, also impact women's reproductive freedom. Reproductive coercion refers to behaviours that interfere with a woman's autonomous decision-making around reproductive health. It includes birth control sabotage, pregnancy coercion, or controlling the outcome of a pregnancy [5]. Women who experience IPV and reproductive coercion often lack reproductive agency and have less control over contraception, sexual intercourse, unplanned or mistimed pregnancy, and partner non-monogamy [6–8]. Gender transformative and gender equity (GE) focused interventions are considered best practices to address these structural factors underlying gender inequality and poor reproductive health outcomes [9]. However, few studies have examined the impact of GE-focused family planning (FP) interventions on reproductive agency indicators, including men's gender-equitable attitudes and women's experiences of IPV and RC. Research on these factors can help us identify the pathways through which GE FP programs can impact women's reproductive health outcomes.

GE-focused and gender-transformative interventions typically examine and disrupt harmful gender norms to reduce gender inequalities and improve health outcomes [9]. These interventions encourage critical reflection on gender norms, promote shared decision-making among couples, and increase women's access to resources. Such programs addressing male engagement in family planning effectively improved reproductive health outcomes in some contexts [10–12]. GE-focused FP programs positively impacted attitudes in studies conducted among a sample of participants in India [13], Guatemala [14] and Kenya [15] but had no impact in Uganda [16] and Malawi [17]. In addition, the pathways through which successful interventions achieve impact are less clear. Even less is known about FP programs' effect on RC. While a growing body of research on RC has documented its negative impact on reproductive health outcomes [7, 18, 19], there is a lack of research on effective models to address RC in family planning interventions in LMICs [19].

The research on IPV is more promising; family planning interventions that include programming on IPV have achieved an impact on IPV. In India, CHARM (Counselling Husbands to Achieve Reproductive health and Marital equity), an FP and GE intervention, significantly improved contraceptive communication, modern contraceptive use among couples, reduced women's experiences of sexual IPV, and improved men's attitudes toward gender norms [13, 20]. Following this successful trial, CHARM was adapted to increase the uptake of female-controlled reversible contraceptive methods and women's participation in the intervention. CHARM2 added two female-only counselling sessions (delivered by female health providers) in parallel to the two male-only sessions provided in the original CHARM intervention, followed by a joint session for the couples [21]. In a cluster randomised controlled trial, CHARM2 increased modern contraceptive use, communication, and contraceptive agency [22]. This study assesses the impact of CHARM2 on GE-related indicators, including IPV, RC, women's and men's attitudes towards physical violence, women's and men's self-reported marital quality, and men's attitudes towards gender norms.

## Methods

### Study design

A two-arm cluster randomised controlled trial was conducted to evaluate CHARM2 (Clinical-Trials.gov #NCT03514914). The intervention occurred between September 2018 and November 2018 and was assessed by interviewing participants at three-time points: baseline (before the intervention) and 9- and 18-month post-intervention between September 2018 and December 2020. The 18-month follow-up was delayed due to the COVID-19 pandemic. The study was implemented in the Junnar block of the Pune district in Maharashtra. This area was chosen due to lower than state-level modern spacing contraceptive use and higher male-to-female sex ratio. In rural Pune, female illiteracy is 27%, and the child sex ratio is 833 girls per 1000 boys (indicative of son preference)" First, the catchment areas of public community health subcentres were chosen as clusters so public health providers could provide the intervention to treatment group participants or routine health and FP care to control group participants. These 20 geographic clusters were then randomised to receive the CHARM2 intervention or the routine standard of care. A sample size of 1200 couples was estimated to have 91.5% power to detect an absolute difference in women reporting IPV of 6% or more between treatment groups at 18-month follow-up (89.2% vs 95.2%); this was based on the effect size difference, control group and within-cluster variance found in the original CHARM data [20].

### Sample

Participants included male-female couples in which women were 18–29 years. Couples were eligible to participate if the participants were not sterilised, had resided in the village together for three months, were planning to stay in the current residence for at least two years, were fluent in Marathi (the local language) and were willing to participate in the study. At baseline, 1201 couples were recruited and provided surveys; couples were recruited between September 10, 2018, and June 20, 2019. At the 9-month follow-up, 1083 couples provided surveys. At the 18-month follow-up, 1087 couples provided surveys. Full-couple retention at 18 months was 90.5% (89.2% intervention, 91.8% control, $p = 0.11$). The full intervention was received by 87.5% of participants, and an additional 7.3% of women and 5.3% of men received the intervention in part (See Fig 1). Further details on the study rationale, setting, methods, sampling, and intervention are available elsewhere [21, 22].

### Procedure and ethical approval

Written consent was obtained from all participants; gender-matched staff obtained consent and conducted interviews with each spouse separately and privately. A couple was eligible only if both spouses consented to the study. Participants were recruited in a two-stage process. First, study staff screened all households in the cluster for couples meeting study eligibility criteria by approaching households face-to-face and compiling a list of all eligible couples. Second, we used a random number generator to determine which couples from the eligible list to approach for enrolment. This enrolment was conducted at the participant's home via face-to-face recruitment. Couples were recruited from their homes and whoever was at home was approached first. If both members of a couple were present, consent was obtained from them simultaneously by gender-matched study staff.

Ethical approval for the CHARM2 study was obtained from the University of California San Diego and the Indian Council of Medical Research-National Institute for Research in Reproductive Health. All research staff and intervention providers were trained to work with

*Ns are couples unless otherwise noted*

40 clusters randomized

20 intervention clusters | 20 control clusters

Screened: 3087

*Excluded at screening (n=395)*
Did not meet inclusion criteria (n=395)
Refused at screening (n=0)

Number of couples screened intervention clusters (n=1649)

Number of couples screened control clusters (n=1438)

*Excluded at screening (n=281)*
Did not meet inclusion criteria (n=281)
Refused at screening (n=0)

Eligible: 2411

Eligible for inclusion in intervention (n=1254)

Eligible for inclusion in control (n=1157)

*Excluded from recruitment (n=374)*
Sufficient recruitment from cluster before couple approached (n=353)
Another couple recruited from same household (n=21)

*Excluded from recruitment (n=281)*
Sufficient recruitment from cluster before couple approached (n=269)
Another couple recruited from same household (n=12)

Approached: 1756

*Did not participate (n=280)*
Refused at baseline (n=91)
Both unavailable at baseline (n=34)
1 partner unavailable at baseline (n=128)
Ineligible upon eligibility reexamination (n=27)

Approached intervention (n=880)

Approached control (n=876)

*Did not participate (n=275)*
Refused at baseline (n=86)
Both unavailable at baseline (n=66)
1 partner unavailable at baseline (n=95)
Ineligible upon eligibility reexamination (n=28)

Enrolled: 1201

Enrolled intervention (n=600)
Received full intervention (n=525)
Received partial intervention (n=57)
Did not receive intervention (n=18)

Enrolled control (n=601)

Surveyed baseline: 1201

*Lost to follow-up 9mo (n=41)*
Moved out of study area (n=13)
Refused: General (n=21)
Refused: Couple separated (n=4)
Refused: Husband died (n=1)
Unable to contact (n=2)

Surveyed baseline intervention (n=600)

Surveyed baseline control (n=601)

*Lost to follow-up 9mo (n=38)*
Moved out of study area (n=13)
Refused: General (n=14)
Refused: Couple separated (n=8)
Refused: Husband died (n=2)
Unable to contact (n=1)

**Fig 1. CHARM2 recruitment and retention consort flow diagram.**

women experiencing IPV. We followed the WHO guidelines for domestic violence research to ensure participants' safety [23].

## Intervention

The CHARM2 intervention involves five sessions of FP and GE counselling delivered over four to six months, with one month between sessions. Gender-matched health providers gave

two sessions separately to husbands and wives (i.e., gender-synchronized). A male or female provider provided a final session for couples (both husband and wife attended this session together), who delivered the individual sessions for either spouse based on availability or participants' preference. Providers included the ANM (Auxiliary Nurse Midwife) attached to the government health subcentre and local private health providers of both genders, allopathic and non-allopathic (ayurvedic and homoeopathic) doctors. CHARM2 providers received a two-day training on GE issues and person-centred FP. Providers in the government health centre also received formal government FP training. All intervention providers used a visual flip chart covering the themes of CHARM2 content and cards with FP methods to facilitate counselling sessions. No incentives were given to ANMs upon the advice of a local government partner as this work was considered within scope of practice and to demonstrate feasibility of scale-up within public health system without additional payment. Two private providers were compensated a nominal salary for 6 months to account for time potentially taken from other paying patients.

CHARM2 FP content for both spouses included exploration of fertility goals and counselling on contraceptive options to achieve these goals. For women, counselling also addressed RC from partners or family. It used a person-centred care approach in which women's choice and fertility goals were central to contraceptive decision-making, including discussing the potential for covert contraceptive use. For men, counselling emphasised the importance of male engagement and respectful communication with wives. Both women and men received input on domestic violence, its causes, reproductive coercion, and steps that can be taken to prevent violence. In addition to the individual sessions, a joint session consolidated the main takeaways of the intervention around methods of family planning and the importance of communication and joint decision-making.

GE elements of the program included dialogue on the importance of respect for women and girls, social norms related to son preference, and the effect male dominance and marital violence can have on healthy and positive marital dynamics and the health of women and children. Counselling sessions could occur at the health provider's office or at the participant's home, as preferred by participants and providers. Most visits took place at health providers' offices. Sessions were 20–40 minutes. More details on the intervention are available elsewhere [21, 22]. Two sessions of the CHARM2 intervention (one session *each* for men and women) had content on sexual violence.

## Measures

**Women only.** *IPV*. CHARM2 used the WHO instrument for assessing IPV [24]. The study included measures of physical IPV (assessed through 6 items), sexual IPV (2 items), and emotional IPV (3 items). All IPV items were evaluated only for female participants and used behaviourally specific questions to inquire about women's experience of acts of IPV (e.g., 'In the past 12 months, has your husband slapped you?'). We created binary measures for each type of violence (e.g., did the respondent report any act of violence in the past year: yes/no). Participants were asked to recall IPV that occurred in the past 12 months at baseline and in the past nine months at the 9-month and 18-month follow-up.

*Reproductive coercion.* We used a 12-item scale to measure reproductive coercion that has been validated in India [7] (e.g., 'Which of the following things has your husband or mother-in-law ever done to oppose your using family planning in the past 12 months? Stopped you from attending a clinic or community 'health day' to obtain family planning?'). We created a binary measure where a participant was coded as having experienced RC if she experienced any one of the twelve acts of RC in the past year.

**Men only.** *Gender-equitable attitudes*. CHARM2 used the Gender-Equitable Men Scale [25] to measure men's attitudes towards gender norms. The scale includes 24 items, such as "a woman should tolerate violence to keep her family together", with response options ranging from 1 (agree) to 3 (do not agree). For this analysis, we used the shorter scale version with 12 items previously validated in the same region due to higher reliability and validity [13]. The item scores were combined to create a mean score. Cronbach's alpha was 0.71. Higher scores indicate more equitable attitudes.

**Women and men.** *Marital quality*. We adopted the Dyadic Adjustment Scale to assess relationship quality among married or cohabiting couples [26]. Marital quality is a predictor of a couple's well-being and is often used interchangeably with terms such as satisfaction, adjustment, well-being, happiness, and success [27]. The original scale had 32 items, and we retained nine for CHARM2. These nine items were tested before the baseline survey and validated using factor analysis [28]. Because items had different response options, each item was rescaled to 0–1, and a mean score was computed. Cronbach's alpha for the final scale was 0.84 for women and 0.63 for men. Higher scores indicate better marital quality.

*Attitudes about physical IPV*. CHARM2 used the National Family and Health Surveys items to measure attitudes towards physical IPV among all participants [1]. The measure included five items (e.g., 'Sometimes a husband is annoyed or angered by things his wife does. In your opinion, is a husband justified in hitting or beating his wife in the following situations: if she goes out without telling him). Cronbach's alpha was 0.79 for women and 0.80 for men. Lower scores indicate less acceptance of violence. This scale was measured as a count variable.

*Poverty*. Below Poverty Line (BPL) cards are issued to families whose household incomes fall under the state-specified poverty threshold. Households use these BPL cards to access subsidized grains, cereals, and sugar under the National Food Security Act. Ownership of BPL cards is used as a proxy for poverty and measured as a binary variable (0 = no, 1 = yes).

*Caste*. Participants were asked to denote their caste, and the caste status variable was recoded as a binary variable to identify participants who belonged to Scheduled Castes, Scheduled Tribes, or Other Backward Classes (0 = no, 1 = yes). SC/ST/OBC individuals are a socioeconomically minoritized group in India.

**Analysis.** We first described the baseline characteristics of women and men, followed by bivariate associations between all outcome measures by treatment group. Next, we used t-tests for continuous variables and Fisher's tests for categorical variables. All outcomes were assessed using an intent-to-treat approach. We used multilevel mixed-effects models to assess differences over time between the intervention and control groups. We used mixed effects logistic regression for binary variables, linear regression for continuous variables, and Poisson regression for count variables. Fixed and random effects were included to account for the study design. The fixed effects terms included the intervention arm, the wave of data collection, and an interaction term for the intervention arm and data collection wave. The random effects terms accounted for individuals nested in subcentres (unit of cluster randomisation). We also included certain variables as fixed effects terms if they were significantly associated with treatment or female loss to follow-up. These include baseline measures of religion, wife living in the same household as her mother-in-law, wife and husband age, wife age at marriage, wife parity, having a living son, scheduled tribe/scheduled caste designation, and poverty status. More details are available in the primary trial paper [22]. We used Stata 17 [29] for data analysis, and all comparisons were assessed at the 5% significance level.

## Results

### Demographics

Table 1 presents the demographic composition of the sample. At baseline, on average, women were 24 years old, and men were 29 years. A large proportion of the sample had completed at least secondary education (86% of women and men). One-third of the sample belonged to SC/ST/OBC minoritized caste groups. Participants had an average monthly household income of Rs. 25,182, and 25% of the sample reported experiencing poverty. At baseline, all men and 54% of women worked for pay in the past year.

There were some significant differences between intervention and control groups at baseline. For example, women in the intervention group were more likely to be Hindu (96% vs. 88%) and live in the same household as their mothers-in-law (82% vs. 78%) than the control group.

### IPV

Ten per cent of women in the intervention group and 13% in the control group reported experiencing physical IPV in the past 12 months at baseline ($p = 0.06$). In multivariate analysis, there was no difference in the experience of physical IPV between the intervention and control groups over time (aOR at 9 months: 1.33, C.I.: 0.60, 2.98; aOR at 18 months: 1.73, C.I.: 0.87, 3.46). These results are presented in Table 2.

At baseline, in the past year, 2% of women in the intervention group and 3% in the control group reported experiencing sexual IPV ($p = 0.46$). In multivariate models, the intervention did not impact sexual violence (aOR at 9 months: 1.01, C.I.: 0.16,6.35; aOR at 18 months: 1.72, C.I.: 0.47,6.38) over time.

Overall, 17% of women in the intervention group and 18% in the control group experienced emotional IPV at baseline in the past 12 months ($p = 0.59$). At 18 months, there was an increase in emotional IPV to 24% in the intervention and 28% in the control group, $p = 0.50$). Both unadjusted and adjusted models found no significant difference between groups on experiences of emotional IPV over time (aOR at nine months: 0.93, C.I.: 0.53,1.63; aOR at 18 months: 0.85, C.I.: 0.53,1.36).

### Gender-equitable attitudes

At baseline, there was no difference among men in their reporting of gender-equitable attitudes, with participants in the intervention group reporting higher gender-equitable attitudes ($M = 2.06$ for intervention, 2.03 for control, $p = 0.17$). However, at 9 months follow-up, men in the intervention group reported significantly more equitable attitudes towards gender norms than men in the control group (2.30 for intervention, 2.14 for control, $p < .001$). At the 18-month follow-up, this difference was sustained (2.51 for intervention, 2.21 for control, $p < .001$). In multivariate analysis, men in the intervention group had a higher likelihood of reporting more gender-equitable attitudes as compared to men in the control group at both the 9- and 18-month follow-up (ß at nine months: 0.13, C.I.: 0.06,0.20; ß at 18 months: 0.26, C.I.: 0.19,0.34).

### Reproductive coercion

At baseline, 8% of women in both the intervention and control groups reported experiencing reproductive coercion in the past 12 months ($p = 0.90$). In multivariate models, there was no difference in reporting of experience of reproductive coercion between the intervention and

**Table 1. Characteristics of CHARM2 participants at baseline, by group (n = 1201).**

| | Overall | Control | Intervention | p-value |
|---|---|---|---|---|
| *N* | 1201 | 601 | 600 | |
| Wife age, mean (SD) | 23.9 (3.0) | 23.9 (3.0) | 23.9 (2.9) | 0.96 |
| Husband age, mean (SD) | 29.4 (3.8) | 29.4 (3.7) | 29.4 (3.9) | 0.8 |
| Wife age at marriage, mean (SD) | 19.4 (2.3) | 19.4 (2.4) | 19.5 (2.3) | 0.55 |
| Child marriage (wife married <18) | | | | 0.5 |
| No | 987 (82.2%) | 489 (81.4%) | 498 (83.0%) | |
| Yes | 214 (17.8%) | 112 (18.6%) | 102 (17.0%) | |
| Couple age difference H-W, mean (SD) | 5.6 (3.2) | 5.5 (3.1) | 5.6 (3.3) | 0.8 |
| Wife parity | | | | 0.31 |
| 0 | 197 (16.4%) | 92 (15.3%) | 105 (17.5%) | |
| 1 | 644 (53.6%) | 324 (53.9%) | 320 (53.3%) | |
| 2 | 315 (26.2%) | 157 (26.1%) | 158 (26.3%) | |
| 3+ | 45 (3.7%) | 28 (4.7%) | 17 (2.8%) | |
| Wife highest education completed | | | | 0.24 |
| Primary or No education (0–8) | 169 (14.1%) | 93 (15.5%) | 76 (12.7%) | |
| Secondary (9–10) | 345 (28.7%) | 163 (27.1%) | 182 (30.3%) | |
| Higher secondary (11–12) | 321 (26.7%) | 169 (28.1%) | 152 (25.3%) | |
| Post-secondary(13+) | 366 (30.5%) | 176 (29.3%) | 190 (31.7%) | |
| Husband highest education completed | | | | 0.94 |
| Primary or No education (0–8) | 174 (14.5%) | 86 (14.3%) | 88 (14.7%) | |
| Secondary (9–10) | 368 (30.6%) | 188 (31.3%) | 180 (30.0%) | |
| Higher secondary (11–12) | 305 (25.4%) | 154 (25.6%) | 151 (25.2%) | |
| Post-secondary(13+) | 354 (29.5%) | 173 (28.8%) | 181 (30.2%) | |
| Wife worked in past year | | | | 0.27 |
| No | 556 (46.3%) | 288 (47.9%) | 268 (44.7%) | |
| Yes | 645 (53.7%) | 313 (52.1%) | 332 (55.3%) | |
| Husband worked in past year | | | | – |
| No | 0 (0%) | 0 (0%) | 0 (0%) | |
| Yes | 1201 (100%) | 601 (100%) | 600 (100%) | |
| Religion | | | | <0.001 |
| Hindu | 1110 (92.4%) | 529 (88.0%) | 581 (96.8%) | |
| Muslim/Buddhist/Jain/Christian/Other | 91 (7.6%) | 72 (12.0%) | 19 (3.2%) | |
| SCST designation | | | | 0.29 |
| None/other | 818 (68.1%) | 418 (69.6%) | 400 (66.7%) | |
| SC/ST/OBC | 383 (31.9%) | 183 (30.4%) | 200 (33.3%) | |
| Husband-reported monthly income Rs, mean (SD) | 25182 (51131) | 27046 (64613) | 23315 (32384) | 0.21 |
| Household has BPL card | | | | 0.84 |
| No | 902 (75.2%) | 453 (75.5%) | 449 (75.0%) | |
| Yes | 297 (24.8%) | 147 (24.5%) | 150 (25.0%) | |
| Has living son | | | | 0.52 |
| No | 645 (53.7%) | 317 (52.7%) | 328 (54.7%) | |
| Yes | 556 (46.3%) | 284 (47.3%) | 272 (45.3%) | |
| Mother-in-law lives in same household | | | | 0.097 |
| No | 240 (20.0%) | 132 (22.0%) | 108 (18.0%) | |
| Yes | 961 (80.0%) | 469 (78.0%) | 492 (82.0%) | |

**Table 2. Multivariate results for CHARM2 participants.**

| Outcome Variables | Arm | Baseline %/Mean | N/SE | 9month follow up %/Mean | N/SE | 18month follow up %/Mean | N/SE | 9-month outcome aOR/aIRR/B | Confidence Intervals Lower | Upper | p | 18-month outcome aOR/aIRR/B | Confidence Intervals Lower | Upper | p |
|---|---|---|---|---|---|---|---|---|---|---|---|---|---|---|---|
| Physical violence (women), % yes | I | 9.5 | 57 | 6.33 | 34 | 7.65 | 41 | 1.33 | 0.60 | 2.98 | 0.48 | 1.73 | 0.87 | 3.46 | 0.12 |
|  | C | 12.98 | 78 | 7.25 | 40 | 7.27 | 40 |  |  |  |  |  |  |  |  |
| Sexual violence (women), % yes | I | 2.17 | 13 | 1.12 | 6 | 1.31 | 7 | 1.01 | 0.16 | 6.35 | 0.99 | 1.72 | 0.47 | 6.38 | 0.41 |
|  | C | 2.83 | 17 | 1.45 | 8 | 0.91 | 5 |  |  |  |  |  |  |  |  |
| Emotional violence (women), % yes | I | 17 | 102 | 15.46 | 83 | 24.07 | 129 | 0.93 | 0.53 | 1.63 | 0.80 | 0.85 | 0.53 | 1.36 | 0.50 |
|  | C | 18.17 | 109 | 17.39 | 96 | 28.08 | 155 |  |  |  |  |  |  |  |  |
| Reproductive coercion (women), % yes | I | 7.69 | 46 | 1.49 | 8 | 3.36 | 18 | 0.43 | 0.16 | 1.19 | 0.11 | 0.85 | 0.25 | 2.88 | 0.79 |
|  | C | 7.50 | 45 | 3.09 | 17 | 3.81 | 21 |  |  |  |  |  |  |  |  |
| Men's gender-equitable attitudes, scale score | I | 2.06 | 0.02 | 2.3 | 0.02 | 2.51 | 0.02 | 0.13 | 0.06 | 0.20 | 0.00 | 0.26 | 0.19 | 0.34 | 0.00 |
|  | C | 2.03 | 0.02 | 2.14 | 0.02 | 2.21 | 0.01 |  |  |  |  |  |  |  |  |
| Attitudes toward physical violence (women), scale score | I | 1.40 | 0.07 | 0.70 | 0.05 | 0.29 | 0.03 | 0.57 | 0.46 | 0.71 | 0.00 | 0.38 | 0.23 | 0.61 | 0.00 |
|  | C | 1.44 | 0.07 | 1.29 | 0.08 | 0.78 | 0.06 |  |  |  |  |  |  |  |  |
| Attitudes toward physical violence (men), scale score | I | 2.1 | 0.08 | 0.82 | 0.05 | 0.20 | 0.03 | 0.64 | 0.45 | 0.90 | 0.01 | 0.25 | 0.18 | 0.39 | 0.00 |
|  | C | 2.30 | 0.08 | 1.46 | 0.07 | 0.91 | 0.05 |  |  |  |  |  |  |  |  |
| Marital quality (women), scale score | I | 0.87 | 0.00 | 0.88 | 0.00 | 0.87 | 0.00 | -0.02 | -0.04 | 0.00 | 0.08 | -0.02 | -0.04 | 0.00 | 0.11 |
|  | C | 0.85 | 0.01 | 0.88 | 0.00 | 0.87 | 0.00 |  |  |  |  |  |  |  |  |
| Marital quality (men), scale score | I | 0.83 | 0.00 | 0.83 | 0.00 | 0.85 | 0.00 | 0.01 | -0.01 | 0.03 | 0.31 | 0.03 | 0.01 | 0.05 | 0.01 |
|  | C | 0.84 | 0.00 | 0.83 | 0.00 | 0.82 | 0.00 |  |  |  |  |  |  |  |  |

*I: Intervention group, C: control group. All models included controls for baseline measures of religion, wife living in the same household as her mother-in-law, wife and husband age, wife age at marriage, wife parity, having a living son, scheduled tribe/scheduled caste designation, and poverty status.

control groups over time (aOR at 9 months: 0.43, C.I.: 0.16,1.19; aOR at 18 months: 0.85, C.I.: 0.25, 2.88).

## Marital quality

In bivariate analysis, among women, at baseline, there was a significant difference between groups in their reports of marital quality ($M = 0.87$ for intervention, 0.85 for control, $p = 0.01$). However, this difference was not sustained at the 9 or 18-month follow-ups. Multivariate analysis showed no intervention impact on marital quality among women (ß at 9 months: -0.02, C.I.: -0.04,0.00; ß at 18 months: -0.02, C.I.: -0.04,0.00) over time.

Among men, at baseline, there was no significant difference between the intervention and control groups in their experience of marital quality ($M = 0.83$ for intervention, 0.84 for control, $p = 0.46$). At the 9-month follow-up, there was no difference between groups in their reports of marital quality. However, at the 18-month follow-up, men in the intervention group reported higher marital quality than men in the control group ($M = 0.85$ for intervention, 0.82

for control, $p<0.001$). Multivariate analysis also showed a significant difference between groups at the endline (ß at 9 months: 0.01, C.I.: -0.01,0.03; ß at 18 months: 0.03, C.I.: 0.01,0.05).

### Attitudes towards physical violence

At baseline, groups reported no difference in attitudes towards physical violence ($M$ = 1.40 for intervention, 1.44 for control, $p$ = 0.71). However, at 9 months follow-up, women in the intervention group were significantly less likely to endorse attitudes justifying physical IPV ($M$ = 0.70 for intervention, 1.29 for control, $p$ = 0.00), and this difference was sustained at 18 months follow-up ($M$ = 0.29 for intervention, 0.79 for control, $p<0.001$). In multivariate models, the intervention was found to have a significant impact on attitudes toward violence at both the 9- and 18-months follow-up periods (aIRR at 9 months: 0.57, C.I.: 0.46,0.71; aIRR at 18 months: 0.38, C.I.: 0.23,0.61).

At baseline, men in the intervention group reported better attitudes towards physical violence ($M$ = 2.10 for intervention, 2.30 for control, $p$ = 0.02). Nine months after the intervention, men in the intervention group reported significantly improved attitudes towards physical violence than men in the control group ($M$ = 0.82 for intervention, 1.46 for control, $p < .001$). This difference was sustained at the 18-month follow-up ($M$ = 0.20 for intervention, 0.91 for control, $p < .001$). Multivariate analyses also found a significant intervention effect for attitudes towards physical violence among men (aIRR at 9 months: 0.64, C.I.: 0.45,0.90; aIRR at 18 months: 0.25, C.I.: 0.18,0.39).

## Discussion

This cluster-randomized trial tested the effectiveness of CHARM2, a gender equity and family planning intervention in rural Maharashtra, over 18 months. Men in the intervention group reported more gender-equitable attitudes at the 9 and 18-month follow-ups and increased marital quality at the 18-month follow-up compared to the control group. Both women and men in the intervention group were also less likely to endorse attitudes justifying physical IPV than the control group. Our results are consistent with other GE and gender-transformative interventions that have positively impacted gender attitudes in India [13, 30]. CHARM2 encouraged participants to reflect on gender inequalities and traditional social norms underlying IPV, RC, and son preference and emphasized the importance of shared decision-making and healthy communication for a happy married life. These factors may have led to a shift in attitudes towards more egalitarian gender norms. CHARM2 provides a template for programming on gender-equitable attitudes that healthcare professionals could implement. Future studies can test the effectiveness of the intervention in improving gender-equitable attitudes in urban areas in India.

Despite improving attitudes towards physical violence, the intervention did not impact women's experiences of male-to-female IPV or reproductive coercion. There are several explanations for this result. One possibility is that additional sessions were needed to translate attitudinal change into behavioural change. The intervention improved participants' attitudes toward physical violence, and prior studies found that norm-based interventions are more likely to succeed with additional programming when there is attitudinal support [31]. Second, CHARM2 included content on sexual violence in only two of five sessions (one session *each* for men and women separately), and a higher dosage may have extended the intervention's impact on behaviours. For instance, a prior intervention with men in India suggests that behavioural change can be a complex process with multiple stages. It took several sessions for men to accept and recognise gender inequitable behaviours and admit they might need to change

[30]. Global reviews of violence prevention interventions also indicate that violence prevention interventions require sustained engagement with participants ranging from 26–50 hours to achieve impact [32–34].

Alternatively, the prevalence rate for women's experiences of IPV in CHARM2 was substantially lower than the state-level IPV prevalence rate. This difference could be attributed to an underreporting bias or a difference in prevalence rates in the study area with a higher socioeconomic status compared to other parts of the state. Due to the low prevalence rates, it could be difficult to see an impact due to a ceiling effect. Still, the intervention may need more intensive programming to achieve an impact within areas with low IPV prevalence rates. It is also possible that the content of the intervention needs to be modified to include explicit discussions of power and privilege underlying gender-based violence and more opportunities for critical reflection to achieve impact on IPV and RC.

COVID-19 may have also contributed to the lack of impact on IPV, RC and marital quality for women. While rates for physical and sexual IPV reduced between baseline, 9 and 18 months follow up, rates for emotional IPV increased between 9 and 18 months follow up when the pandemic started. This increase occurred in both intervention and control groups, suggesting that an external factor like the pandemic may have contributed to increased emotional IPV. Studies have reported increased IPV in India in the aftermath of the COVID-19 pandemic owing to illness-related stressors, home confinement, pre-existing socioeconomic vulnerabilities, increased burden of unpaid labour, limited mobility, and financial stressors [35, 36]. These factors could have increased marital conflict leading to higher emotional IPV and lower marital quality for women.

While CHARM2 positively impacted men's marital quality, we do not see a similar effect on women's marital quality. Men and women may have different views of their relationships. While primary trial analyses indicate that the intervention helped couples improve their communication skills, it is possible that this change impacted men more than women. Future studies can assess differences in the relevance of different relationship factors for men and women. Additionally, this study relied on women's reports of male-to-female IPV, and men were not asked about their perpetration. IPV has been linked with lower marital quality, and gender differences in understanding and reporting of IPV may partially explain the difference in marital quality between women and men [28, 37, 38].

The findings of this study should be interpreted considering its limitations. Our study's prevalence of IPV was low compared to the prevalence reported in country-wide surveys. This difference may be partially attributable to our sample of young married couples. Prior research has found that older women report higher levels of IPV [39, 40]. There may also have been some selection bias where couples reporting lower levels of IPV chose to participate in the study. However, this is unlikely as there was no difference in reporting rates of IPV at baseline between the intervention and control groups. All measures relied on self-report and are therefore subject to disclosure and underreporting bias. We attempted to mitigate social desirability bias by using researchers who were external to the program, trained in gender sensitivity and emphasised the confidentiality of all answers.

## Conclusion

This study demonstrates that the CHARM2 GE and FP couples' intervention effectively improved male and female attitudes towards physical IPV and men's gender-equitable attitudes and perceptions of marital quality. However, the intervention did not impact women's experiences of IPV or reproductive coercion. This may be partially attributed to various factors, including the low prevalence of IPV leading to a ceiling effect, insufficient dosage and

content on IPV, and the stressors associated with the COVID-19 pandemic. Nonetheless, the programme demonstrates that the FP GE approach can be used safely with couples in India to transform attitudes towards IPV and gender norms.

## Supporting information

**S1 Checklist. Inclusivity in global research.**
(DOCX)

**S1 File. Analysis dataset in STATA format.**
(DTA)

**S2 File. Analysis dataset in.xlx format.**
(XLSX)

**S3 File. STATA code for analysis.**
(DO)

## Author Contributions

**Conceptualization:** Sangeeta Chatterji, Anita Raj.

**Data curation:** Sangeeta Chatterji, Nicole E. Johns.

**Formal analysis:** Sangeeta Chatterji, Nicole E. Johns.

**Funding acquisition:** Anita Raj.

**Investigation:** Sarah Averbach, Anita Raj.

**Methodology:** Anita Raj.

**Project administration:** Mohan Ghule, Shahina Begum.

**Writing – original draft:** Sangeeta Chatterji.

**Writing – review & editing:** Nicole E. Johns, Mohan Ghule, Shahina Begum, Sarah Averbach, Madhusudana Battala, Anita Raj.

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
