## [Decision Letter · Decision Letter 0]

31 Jan 2024

PGPH-D-23-01761

Assessing the impact of CHARM2, a family planning program, on gender attitudes, intimate partner violence, reproductive coercion, and martial quality in India

Dear Dr. Chatterji,

Thank you for submitting your manuscript to PLOS Global Public Health. After careful consideration, we feel that it has merit but does not fully meet PLOS Global Public Health’s publication criteria as it currently stands. Therefore, we invite you to submit a revised version of the manuscript that addresses the points raised during the review process.

We look forward to receiving your revised manuscript.

Kind regards,

Adriana Biney

Academic Editor

Journal Requirements:

2. Please include a complete copy of PLOS’ questionnaire on inclusivity in global research in your revised manuscript. Our policy for research in this area aims to improve transparency in the reporting of research performed outside of researchers’ own country or community. The policy applies to researchers who have travelled to a different country to conduct research, research with Indigenous populations or their lands, and research on cultural artefacts. The questionnaire can also be requested at the journal’s discretion for any other submissions, even if these conditions are not met.  Please find more information on the policy and a link to download a blank copy of the questionnaire here: https://journals.plos.org/globalpublichealth/s/best-practices-in-research-reporting. Please upload a completed version of your questionnaire as Supporting Information when you resubmit your manuscript.

3. Please provide separate figure files in .tif or .eps format only and remove any figures embedded in your manuscript file. Please also ensure all files are under our size limit of 10MB.

4. Please amend your detailed Financial Disclosure statement. This is published with the article. It must therefore be completed in full sentences and contain the exact wording you wish to be published.

a. State what role the funders took in the study. If the funders had no role in your study, please state: “The funders had no role in study design, data collection and analysis, decision to publish, or preparation of the manuscript.”

b. If any authors received a salary from any of your funders, please state which authors and which funders.

If you did not receive any funding for this study, please simply state: “The authors received no specific funding for this work.

5. In the online submission form, you indicated that "The datasets used is available from the last author on reasonable request". All PLOS journals now require all data underlying the findings described in their manuscript to be freely available to other researchers, either 1. In a public repository, 2. Within the manuscript itself, or 3. Uploaded as supplementary information.

Additional Editor Comments (if provided):

Reviewers' comments:

Reviewer's Responses to Questions

**Comments to the Author**

1. Does this manuscript meet PLOS Global Public Health’s publication criteria? Is the manuscript technically sound, and do the data support the conclusions? The manuscript must describe methodologically and ethically rigorous research with conclusions that are appropriately drawn based on the data presented.

Reviewer #1: Yes

Reviewer #2: Yes

Reviewer #3: Yes

2. Has the statistical analysis been performed appropriately and rigorously?

Reviewer #1: Yes

Reviewer #2: Yes

Reviewer #3: Yes

3. Have the authors made all data underlying the findings in their manuscript fully available (please refer to the Data Availability Statement at the start of the manuscript PDF file)?

Reviewer #1: Yes

Reviewer #2: No

Reviewer #3: No

4. Is the manuscript presented in an intelligible fashion and written in standard English?

Reviewer #1: Yes

Reviewer #2: Yes

Reviewer #3: Yes

5. Review Comments to the Author

Reviewer #1: • The main claim of the paper was that a family planning intervention i.e. the CHARM2 intervention, could improve the attitudes towards gender equity and physical Intimate Partner’s Violence. This is an important area which needs cultural shift to promote gender equitable attitudes in rural areas of India.

• The claims are properly placed in the context of the previous literature, they treated the literature fairly and they referred to three other published articles written by same team that contain further detailed information about this study.

• Data were analyzed statistically. Data analyses were relevant and consistent to support the claims. However, it would be good to check the tools by a statistician to confirm their sufficiency and whether other evidence is required.

• As the PLOS Global Public Health encourages authors to publish detailed protocols they published the study protocol previously. Supporting data were ensured to be available on request.

• The paper is considered suitable for publication but reporting the study findings following the CONSORT checklist or justifying how the 25 items of the CONSORT checklist are considered is highly recommended.

• The study include a CONSORT flow diagram to report the recruitment and retention of the study participation.

• The methodology has been sufficiently detailed to allow the experiments to be reproduced.

• The manuscript is well organized and written clearly enough to be accessible to non-specialists but two things can improve the clarity further. These are as follows:

a. Acronyms should be expanded in words/ phrases when they are used for the first time

b. It is highly recommended that the author should justify how the 25 items of the CONSORT checklist are considered in the reporting of the study methods and findings.

Reviewer #2: Dear authors,

Thank you for this interesting manuscript “Assessing the impact of CHARM2, a family planning program, on gender attitudes, intimate partner violence, reproductive coercion, and martial quality in India”. I enjoyed reading it. Below you will find my comments, please follow-up on these to improve the manuscript.

In the methods you refer a couple of times to other articles (eg “Additional details

are available in the study protocol (20, 21)” and “More details on the intervention are available elsewhere (20, 21)”). I advise you to elaborate more on your methods in this manuscript. You cannot expect readers to return to your earlier published articles to review the methods. This paper should be complete and clear.

P5 line 115-116: I understand that the clusters were randomized, so the couples were included depending on the cluster where they were living. This might give bias which you should describe in the limitations.- and response bias which should be described in the limitations.

P5 line 126-127: how were the participating couples recruited from the included clusters? Were couples selected by the HCP and face-to-face asked to participate? Was every couple from sept 10 screened for inclusion by an independent recruiter, or could the treating HCP preselect risking selection bias? Please elaborate on this.

P6 Line 148-150: not clear.

*Was the final session with the couple, or was the final session individual? Or did this depend on the preference of the participating couple? Then this should be described in the results.

*Please describe the aim of this final session with the couple.

*If this final session was done with the couple, response bias will be a possible limitation since people might not dare to speak freely and give desirable answers, so this should be described in the limitations.

P7 line 165-166: what was the reason not to include RC in the counselling of men?

Page numbering (see table 2): the page numbering restarts at 1. Please adjust this to continuous numbering.

Line 307: you write 12 months F/U, but this should be 18 months F/U. Please adjust.

Line 386 and further: limitations

The risk of response bias should be described.

Please add an extra heading “Recommendations for future studies” after the discussion and conclusion (which is line 396-403). You could for example describe a study on behavioral change since this did not change in your study, as pointed at in line 353.

Please make sure that your datasets are part of the manuscript.

Reviewer #3: “Assessing the impact of CHARM2, a family planning program on gender attitudes, IPV, reproductive coercion and marital quality in India” considers an important subject on men’s and women’s wellbeing in the Pune district in Maharashtra. The data is unique and appears rich and the manuscript is fairly clearly written. This was an interesting paper on an important topic. However, I have a few suggestions/comments for the authors.

1. It appears that there are about four outcome variables here and I would think the data used here could have provided two manuscripts. This would have allowed the authors to thoroughly engage the results more.

2. It will be important to explain the terms as used in the paper. For example, what is reproductive coercion, marital quality and gender transformative and gender equity. This allows the reader to understand from the onset what the terms seek to explain or measure.

3. Can you provide a little more information about the research area? Apart from the fact that Maharashtra was chosen due to lower than state level modern contraceptive use and higher male-female sex ratio? For instance, what are traditional gender norms/attitudes/socialization/ women’s empowerment/ women’s level of education in the area and how will these influence the uptake of the intervention. In doing this you are able to add more meaning to the results during the discussion.

4. Line 80 to 81 “…gender equitable attitudes in studies conducted among a sample of participants in India (12)…..” rather than “….gender equitable attitudes in India (12)…..”

5. Line 90-91 “…significantly improved contraceptive communication, modern contraceptive use among couples, REDUCED women’s experiences of sexual IPV and IMPROVED men’s equitable towards gender norms.”

6. It would be important to throw some light on why CHARM reduced women’s experience of IPV but in CHARM2 the intervention did not appear to impact sexual IPV experience. Were there more educational sessions in CHARM than CHARM2. If there were why were the number of educational sessions reduced in CHARM2? Was it just about sample selection? Or something else?

7. Line 93 “……reversible contraceptives methods and women participation” Is it women’s participation in the intervention?

8. Line 97….please clarify why a measure of marital quality was included to assess any potential “BACKLASH”

9. Line 138 “If both PARTNERS were present, consent was obtained from THEM…..” rather than “If both members of a couple were present, consent was obtained from both partners..”

10. Line 220 to 221……. What about others who do not belong to the caste stated here? Could you write something about them? For example, are they economically more empowered? How are they referred to?

11. Line 248 “….between intervention and CONTROL GROUPS at baseline.” rather than “…bet

---

## [Decision Letter · Decision Letter 1]

19 Apr 2024

Assessing the impact of CHARM2, a family planning program on gender attitudes, intimate partner violence, reproductive coercion, and martial quality in India

PGPH-D-23-01761R1

Dear Dr. Chatterji,

We are pleased to inform you that your manuscript 'Assessing the impact of CHARM2, a family planning program on gender attitudes, intimate partner violence, reproductive coercion, and martial quality in India' has been provisionally accepted for publication in PLOS Global Public Health.

Best regards,

Adriana Andrea Ewurabena Biney

Academic Editor

Reviewer Comments (if any, and for reference):

Reviewer's Responses to Questions

**Comments to the Author**

1. If the authors have adequately addressed your comments raised in a previous round of review and you feel that this manuscript is now acceptable for publication, you may indicate that here to bypass the “Comments to the Author” section, enter your conflict of interest statement in the “Confidential to Editor” section, and submit your "Accept" recommendation.

Reviewer #3: All comments have been addressed

2. Does this manuscript meet PLOS Global Public Health’s publication criteria? Is the manuscript technically sound, and do the data support the conclusions? The manuscript must describe methodologically and ethically rigorous research with conclusions that are appropriately drawn based on the data presented.

Reviewer #3: Yes

3. Has the statistical analysis been performed appropriately and rigorously?

Reviewer #3: Yes

4. Have the authors made all data underlying the findings in their manuscript fully available (please refer to the Data Availability Statement at the start of the manuscript PDF file)?

Reviewer #3: Yes

5. Is the manuscript presented in an intelligible fashion and written in standard English?

Reviewer #3: Yes

6. Review Comments to the Author

Reviewer #3: The authors have addressed the comments satisfactorily.

Just two more comments:

1. If the authors have a citation for the statement below Line 116-117 they could include that.

“In rural Pune, female illiteracy is 27%, and the child sex ratio is 833 girls 117 per 1000 boys (indicative of son preference)”

2. A few of the headings could be formatted so that they appear on the next page:

Method

Men only

Results

7. PLOS authors have the option to publish the peer review history of their article (what does this mean?). If published, this will include your full peer review and any attached files.

**Do you want your identity to be public for this peer review?** For information about this choice, including consent withdrawal, please see our Privacy Policy.

Reviewer #3: No
